# When research is me-search: How researchers' motivation to pursue a topic affects laypeople's trust in science

**Marlene Sophie Altenmüller** ⓘ *, **Leonie Lucia Lange, Mario Gollwitzer** ⓘ

Department of Psychology, Ludwig-Maximilians-Universität München, Munich, Germany

* marlene.altenmueller@psy.lmu.de

## Abstract

Research is often fueled by researchers' scientific, but also their personal interests: Sometimes, researchers decide to pursue a specific research question because the answer to that question is idiosyncratically relevant for themselves: Such "me-search" may not only affect the quality of research, but also how it is perceived by the general public. In two studies (N = 621), we investigate the circumstances under which learning about a researcher's "me-search" increases or decreases laypeople's ascriptions of trustworthiness and credibility to the respective researcher. Results suggest that participants' own preexisting attitudes towards the research topic moderate the effects of "me-search" substantially: When participants hold favorable attitudes towards the research topic (i.e., LGBTQ or veganism), "me-searchers" were perceived as more trustworthy and their research was perceived as more credible. This pattern was reversed when participants held unfavorable attitudes towards the research topic. Study 2 furthermore shows that trustworthiness and credibility perceptions generalize to evaluations of the entire field of research. Implications for future research and practice are discussed.

## Introduction

"Being a scientist is, at the most fundamental level, about being able to study what's exciting to you", says Jeremy Yoder, a gay man studying experiences of queer individuals in science [1]. Like Yoder, many researchers are passionate about their research and dedicated to their field. After all, they are free to choose research questions they deem important and are interested in. Freedom of science and research secures the independence of the academic from the political and other spheres. In return, researchers are expected to be neutral and objective and make their research process transparent to guarantee that this freedom is not exploited for personal gains.

Just as people differ in what they are interested in in their personal lives, researchers differ in what they find more or less fascinating and worth studying. Such fascination can have multiple causes and is often rooted in a perceived personal connection to a topic. For instance, Sir Isaac Newton allegedly became interested in gravity after an apple fell on his head [2]. A specific type of personal connection exists when researchers study a phenomenon because they

**Data Availability Statement:** We provided all materials, the anonymized data and analyses, and supplementary materials online at the Open

Science Framework via the following link: https://osf.io/phfq3/.

**Funding:** The author(s) received no specific funding for this work.

**Competing interests:** The authors have declared that no competing interests exist.

are directly (negatively) affected by that phenomenon. In 1996, Harvard alumni and neuro-anatomist Jill Bolte Taylor suffered a rare form of stroke that made her undergo major brain surgery, affected her personal and academic life tremendously, and eventually awakened her interest in studying the plasticity of the brain [3]. In 2006, she published an award-winning book covering her research and her personal story that led her to pursue this path. The Jill Bolte Taylor case is, thus, a prototypical example for such "me-search": researchers studying a phenomenon out of a particular personal affection by (or connection to) this phenomenon. "Me-search" thus means pursuing a scientific question when the answer to that question is idiosyncratically relevant for the individual researcher (as opposed to when the answer is relevant per se).

Being directly affected by a phenomenon provides researchers studying it with a high degree of expertise and motivation: Jill Bolte Taylor, for instance, claims to bring a deep personal understanding and compassion to her research and work with patients [4, 5]. That said, being personally affected may also come at the cost of losing one's scientific impartiality and neutrality for the subject: Jill Bolte Taylor was criticized for being overly simplistic in her scientific claims and mixing them with esoteric ideas, and for pushing her own agenda (i.e., selling her story) by dramatizing her own experiences [4–7].

While some criticized Jill Bolte Taylor heavily, the general public does not seem to have a problem with her research as "me-search". Her book is currently translated into 30 languages, and thousands of people visit her talks and keynote addresses [4–6]. Does that suggest that the general public tends to turn a blind eye on conflicts of interest that may arise from a researchers' personal affection by their research object? While the Jill Bolte Taylor case seems to suggest so, research on science communication and public understanding of science has shown that people are highly sensitive to potential conflicts of interest arising from researchers' personal involvement: perceiving researchers as pursuing an "agenda" for personal reasons is a major factor predicting people's loss of trust in researchers and science [8–11]. On the other hand, people may see personal ("autoethnographic") experiences of researchers personally affected by their topic as valuable and laudable – it may imply that "they know what they're talking about" [12–14]. Similarly, revealing a personal interest or even passion for a particular research topic (e.g., due to being personally affected) could also overcome the stereotypical perception of scientists as distant "nerds in the ivory tower" [15, 16]: researchers who openly disclose the idiosyncratic relevance of their research topic may appear more approachable, more likeable, and more trustworthy [17–19].

Thus, the public's reaction to "me-search" seems to be ambivalent and contingent on certain boundary conditions. Thus, the question we are going to address in this article is whether and when – that is, under which circumstances – a researcher's personal affection by a research topic ("me-search") positively vs. negatively impacts public perceptions regarding the *trustworthiness* of the respective researcher (and the entire research area in general) and the extent to which this researcher's findings are perceived as *credible*.

## Perceivers' motivated stance as a moderating variable

This potentially ambivalent perception of "research as me-search" can be understood from a *motivated reasoning* [20] perspective: Laypeople receive and process information in a manner biased towards their own beliefs, expectations, or hopes. This also applies to the reception of scientific information [21, 22]: For example, laypeople are more likely to dismiss scientific evidence if it is inconsistent with their beliefs [23, 24] or if it threatens important (moral) values [25, 26] or their social identity, respectively [27–29].

However, identity-related and attitudinal motivated science reception might differ in their underlying mechanisms. For identity-related motivated science reception, biased perception of information, which is relevant to a social identity, is driven by a defense motivation to protect this positive social identity [30]. Thus, identity-threatening scientific information is countered by identity-protection efforts, such as discrediting the findings and the source. These efforts will be more pronounced among strongly identified individuals [27–29]. For attitudinal motivated science reception, however, the mechanism might function as a broader perception filter. When confronted with new scientific information about the respective attitude object, the perceptual focus will be directed at clues helping to uphold prior attitudes (i.e., *confirmation bias* [31]): Potentially attitude-inconsistent information is attenuated, while potentially attitude-consistent information is accentuated. The ambivalent nature of "me-search" might allow to be easily bend in such a motivated manner and, thus, lead to biased perceptions of a researcher either way: when the findings are in line with one's prior beliefs, being personally affected may be considered an asset–the respective researcher is perceived as more trustworthy and his/her findings as more credible (compared to no idiosyncratic relevance). However, when the findings are inconsistent with one's prior beliefs, idiosyncratic relevance may be considered a flaw–the respective research is perceived as biased, untrustworthy, and less competent, and his/her findings are likely perceived as less credible than when idiosyncratic relevance is absent.

Prior research on motivated science reception mainly focused on laypeople's reactions towards specific scientific findings: *after* learning about the outcome of a particular study, participants dismiss the research (and devalue the researcher) if these outcomes are consistent vs. inconsistent with their prior beliefs [23–25, 27–29]. However, people might be prone to motivated science reception even *before* results are known, judging researchers proverbially just by their cover (e.g., by biographical data, personal and scientific interests and motivations). People who hold positive attitudes towards a certain research topic might perceive "me-searchers" as more trustworthy and anticipate their results to be more credible (before knowing the specific outcomes). By contrast, people who hold negative attitudes towards a certain research topic they might trust "me-searchers" less and expect their findings to be less credible.

Additionally, motivated reception processes can be extended over and above the specific information under scrutiny and lead to questioning the scientific method in itself–a phenomenon termed the "scientific impotence excuse" [32]. In line with that, critical evaluations of specific researchers and their findings are sometimes generalized to the entire field of research [27]. Thus, the fact that a researcher engages in "me-search" might be interpreted in a way that fits best to one's preexisting convictions and may generalize to the entire field of research.

## The present research

In two studies, laypeople read alleged research proposals concerning potentially polarizing research topics (i.e., LGBTQ issues and veganism) which were submitted by researchers who disclosed being either personally affected or not affected by the respective topic. We investigated whether (Study 1) and when (i.e., moderated by preexisting positive attitudes towards the respective research topic, Studies 1 and 2) such "me-search" information increased or decreased laypeople's perceptions regarding these researchers' epistemic trustworthiness and the anticipated credibility of their future scientific findings. Of note, we use the term "credibility" to differentiate evidence-related trust/credibility from person-related trust/credibility (i.e. "trustworthiness"). Further, we test whether one researcher's "me-search" impacts the evaluation of the entire respective field (Study 2).

For both studies in this paper, we report how we determined our sample size, all data exclusions (if any), all manipulations, and all measures [33]. All materials, the anonymized data, and analyses are available online at the Open Science Framework (OSF; see https://osf.io/

phfq3/). Before starting the respective study, informed consent was obtained. Participants read a GDPR-consistent data protection and privacy disclosure declaration specifically designed for the present study. Only participants who gave their consent could start the respective survey. According to German laws and ethical regulations for psychological research [34], gathering IRB approval is not necessary if (i) the data are fully anonymized, (ii) the study does not involve deception, (iii) participants' rights (e.g., voluntary participation, the right to withdraw their data, etc.) are fully preserved, and (iv) participating in the study is unlikely to cause harm, stress, or negative affect. The present studies met all of these criteria; therefore, no IRB approval had to be obtained.

## Study 1

In our first study, we conducted an online experiment investigating the main effect of a researcher's disclosure of being personally affected vs. not affected by their research on their trustworthiness and the credibility of their future research. Further, we tested whether laypeople's preexisting attitudes towards the research topic moderate this effect.

### Method

**Sample.**  Four-hundred and eleven German participants were recruited via mailing lists and social networks. Ninety-seven participants had to be excluded due to pre-specified criteria: Sixty-seven participants failed the manipulation check; 25 participants failed the pre-specified time criteria (viewing the manipulation stimulus less than 30sec, taking less than 3min or more than 20min for participation); 5 participants had apparently implausible response patterns (e.g., "straight-lining;" identical responses on every single item on more than one questionnaire page in a row). Eighty-five further participants failed the attention check. Excluding them did not change the overall results, so, for the sake of statistical power, we did not exclude these 85 cases. The final sample consisted of $N$ = 314 participants. We conducted sensitivity analyses using G*Power [35] for determining which effect sizes can detected with this sample in a moderated (multiple) regression analysis: At $\alpha$ = 0.05 and with a power of 80%, small-to-medium effects ($f^2 \geq 0.03$) can be detected with this sample. Participants were mostly female (74% female, 25% male, 2% other) and their age ranged between 16 and 68 years ($M$ = 26.79; $SD$ = 10.18). Most participants were currently studying at a university (71%; working: 21%; unemployed or other: 8%). Participants who were currently studying or already had a university degree (93%) came from a variety of disciplines (law, economics, and social sciences: 49%; humanities: 16%; mathematics and natural sciences: 14%; medicine and life sciences: 11%; engineering: 4%).

**Materials and procedure.**  After obtaining informed consent, we asked participants to imagine they were browsing the website of a research institute and came across a short proposal for a new research project by a researcher named Dr. Lohr (no gender was indicated for greater generalizability and avoiding possible gender confounds). Next, participants read the beginning of the alleged proposal of a planned research project for which Dr. Lohr was allegedly applying for external funding. The text briefly introduced the planned project (i.e., investigating social reactions to queer employees at the workplace) and a statement of Dr. Lohr explaining why they were interested in conducting this project. Participants were randomly allocated to two groups. In the "not personally affected" condition, Dr. Lohr wrote:

> "*I am interested in investigating this research topic in more detail not only out of scientific reasons but also because I–as someone who does not identify as homosexual and is not affected by my own research–really think we need more evidence-based knowledge about queer topics which we can implement in everyday life.*"

In the "personally affected" condition, Dr. Lohr wrote:

"*I am interested in investigating this research topic in more detail not only out of scientific reasons but also because I–as someone who identifies as homosexual and is affected by my own research–really think we need more evidence-based knowledge about queer topics which we can implement in everyday life.*"

We added a definition for the word "queer" below the proposal: "*Queer is a term used as self-description by people who do not identify as heterosexual and/or who do not identify with the gender assigned at birth. The term is often used as umbrella term for LGBTQ (lesbian, gay, bisexual, trans and queer) and describes all people who identify as queer.*" After completing an attention check question (see pre-registration), we measured participants' trust in Dr. Lohr with the *Muenster Epistemic Trustworthiness Inventory* (METI; [36]), which was constructed for measuring trust in experts encountered online. It consists of 14 opposite adjective pairs measuring an overall trustworthiness score (Cronbach's $\alpha$ = .95) as well as the sub-dimensions expertise (e.g., competent–incompetent, Cronbach's $\alpha$ = .92) and integrity/benevolence (e.g., honest–dishonest, Cronbach's $\alpha$ = .93) on 6-point bipolar Likert scales. Factor analyses (see Appendix A in the supplementary materials, https://osf.io/phfq3/) suggest that a two-factor model (with expertise and integrity/benevolence) fit the data better than a three-factor model (as suggested by [36]), corroborating the idea of a cognitive-rational dimension and an affective dimension of trustworthiness [37]. Next, participants rated the extent to which they found Dr. Lohr's research credible on a 6-point Likert scale ranging from 1 = "not at all" to 6 = "very much" (6 items, e.g., "I think Dr. Lohr's future findings will be credible;" "I will be critical of Dr. Lohr's research results" (reverse-coded); Cronbach's $\alpha$ = .84).

Next, we measured participants' own positive attitudes towards LGBTQ issues—the moderator variable in our design—with eleven statements developed from research on sympathy, group attitudes, and allyship [38, 39] rated on a 6-point Likert scale ranging from "not at all" to "very much" (e.g., "I think that LGBTQ-related topics receive more attention than necessary" (reverse-coded); "I am open to learning more about concerns raised by LGBTQ people;" Cronbach's $\alpha$ = .93). Next, we conducted a manipulation check by asking participants to indicate whether Dr. Lohr disclosed being personally affected by their research ("Dr. Lohr stated being personally affected;" "Dr. Lohr stated not being personally affected;" "Dr. Lohr did not say anything about being affected or not").

Finally, we measured demographic variables (age, gender, occupation, academic discipline) and control variables: general perceptions of researchers' neutrality (self-developed 6-point bipolar scale with 4 adjective-pairs, e.g. subjective–objective, and 6 distractor pairs, e.g. introverted–extraverted, Cronbach's $\alpha$ = .81) and *Public Engagement with Science* (PES) with two measures adapted from a survey by the BBVA Foundation [40]: a 5-item scale measuring *PES frequency* (e.g., "How often do you read news about science?" 5-point Likert scale ranging from 0 ="never" to 5 ="almost daily," Cronbach's $\alpha$ = .78) and a multiple choice question measuring 15 potential *PES experiences* during the last 12 months (e.g., "I know someone who does scientific research;" "I visited a science museum"). Participants had the opportunity to participate in a lottery and sign up for more information and were debriefed.

## Results

Our randomized groups did not differ in regard to general perception of neutrality in science ($p$ = .924) or PES (PES frequency, $p$ = .709; PES experiences, $p$ = .533). Table 1 summarizes all means, standard deviations, correlations and internal consistencies of the measured variables.

**Table 1. Means, standard deviations, correlations and internal consistencies.**

| Variable | M | SD | α | 1 | 2 | 3 | 4 | 5 | 6 | 7 |
|---|---|---|---|---|---|---|---|---|---|---|
| 1. Expertise | 4.61 | 0.91 | .92 | | | | | | | |
| 2. Integrity/benevolence | 4.94 | 0.79 | .93 | .72*** | | | | | | |
| 3. Epistemic trustworthiness | 4.80 | 0.79 | .95 | .92*** | .94*** | | | | | |
| 4. Credibility | 4.10 | 0.92 | .84 | .68*** | .68*** | .73*** | | | | |
| 5. Attitudes towards LGBTQ | 4.93 | 1.02 | .93 | .23*** | .32*** | .30*** | .47*** | | | |
| 6. Neutrality expectation | 4.12 | 0.88 | .81 | .17** | .17** | .18** | .10 | .08 | | |
| 7. PES frequency | 3.25 | 0.71 | .78 | -.08 | -.09 | -.09 | -.08 | -.07 | .10 | |
| 8. PES experiences | 6.92 | 2.98 | - | -.11* | -.05 | -.09 | -.02 | -.03 | .05 | .56*** |

*Note.* N = 314.

* indicates $p < .05$.

** indicates $p < .01$.

*** indicates $p < .001$. α represents internal consistencies (Cronbach's α). Variables 1–6 ranged from 1 to 6, variable 7 ranged from 1–5 and variable 8 ranged from 0–15.

**Main effect of being personally affected.** First, we tested the main effect of the researcher's disclosure of being personally affected on epistemic trustworthiness and credibility of future findings. Laypeople trusted Dr. Lohr significantly more in the "personally affected" condition (M = 4.92, SD = 0.75) than in the "not personally affected" condition (M = 4.66, SD = 0.81), t(312) = 2.93, p = .004, d = 0.33, 95% CI$_d$ [0.11; 0.56]. For credibility, the difference between the "personally affected" condition (M = 4.15, SD = 0.96) and the "not personally affected" condition (M = 4.04, SD = 0.86) was not significant, t(312) = 1.02, p = .306, d = 0.12, 95% CI$_d$ [-0.11; 0.34]. Further exploring the two dimensions of epistemic trustworthiness, Dr. Lohr was perceived as higher on integrity/benevolence, t(312) = 3.19, p = .002, d = 0.36, 95% CI$_d$ [0.14; 0.59], and on expertise, t(312) = 2.17, p = .030, d = 0.25, 95% CI$_d$ [0.02; 0.47] when disclosing being personally affected.

**Moderation by pre-existing attitudes.** Second, we tested whether the effect of being personally affected by the research topic on trustworthiness was moderated by participants' pre-existing attitudes towards LGBTQ issues. Using standardized linear regression, we again found a main effect of condition on trustworthiness, beta = 0.15, p = .004, 95% CI$_{beta}$ [0.05, 0.26]. There was a significant main effect of participants' pre-existing attitudes, beta = 0.30, p < .001, 95% CI$_{beta}$ [0.20, 0.40] and the condition × attitudes interaction effect was significant, beta = 0.19, p < .001, 95% CI$_{beta}$ [0.08, 0.29], increasing the amount of explained variance in trustworthiness by 3% to $R^2_{adj}$ = .14. Table 2 summarizes the results. Fig 1A displays the interaction effect and standardized simple slopes analysis further qualifies it: Participants with more positive attitudes towards LGBTQ issues (+1 SD above sample mean) trusted Dr. Lohr more when the researcher was personally affected vs. not affected, beta = 0.34, p < .001, 95% CI$_{beta}$ [0.20, 0.49]. For participants with less positive attitudes towards LGBTQ issues (-1 SD below sample mean), this effect appears to be reversed, yet the simple slope was not significant, beta = -0.03, p = .646, 95% CI$_B$ [-0.18, 0.11]. The same pattern of interaction effects emerged for both, integrity/benevolence (p = .009, total $R^2_{adj}$ = .14) and expertise (p < .001, total $R^2_{adj}$ = .10); full analyses are reported in Appendix B (see https://osf.io/phfq3/).

Regarding our second dependent variable, credibility, we found no main effect of condition, beta = 0.04, p = .456, 95% CI$_{beta}$ [-0.06, 0.13]. However, there was a significant main effect of participants' pre-existing attitudes, beta = 0.48, p < .001 95% CI$_{beta}$ [0.39, 0.58]: Participants with more positive attitudes anticipated a higher credibility of future research findings in this

**Table 2. Standardized regression results and semi-partial correlations, Study 1.**

| Predictor | beta | beta 95% CI | $sr^2$ | $sr^2$ 95% CI |
|---|---|---|---|---|
| Epistemic trustworthiness | | | | |
| condition | 0.15** | [0.05, 0.26] | .02 | [-.01, .05] |
| attitudes | 0.30*** | [0.20, 0.40] | .09 | [.03, .15] |
| condition × attitudes | 0.19*** | [0.08, 0.29] | .04 | [-.00, .07] |
| Credibility | | | | |
| condition | 0.04 | [-0.06, 0.13] | .00 | [-.01, .01] |
| attitudes | 0.48*** | [0.39, 0.58] | .24 | [.16, .32] |
| condition × attitudes | 0.21*** | [0.12, 0.31] | .05 | [.01, .09] |

*Note.* N = 314.

** indicates $p < .01$.

*** indicates $p < .001$. Condition is standardized by effect coding (-1 = not personally affected, 1 = personally affected). *beta* represents standardized regression weights. $sr^2$ represents the semi-partial correlation squared.

condition than participants with less positive attitudes. Similar to epistemic trustworthiness, there was a significant condition × attitudes interaction effect, *beta* = 0.21, $p < .001$, 95% CI$_{beta}$ [0.12, 0.31], increasing the amount of explained variance in credibility by 4% to $R^2_{adj} = .26$. Table 2 summarizes the results. Fig 1B displays this interaction effect: Again, participants with more positive attitudes towards LGBTQ issues (+1 *SD* above sample mean) anticipated Dr.

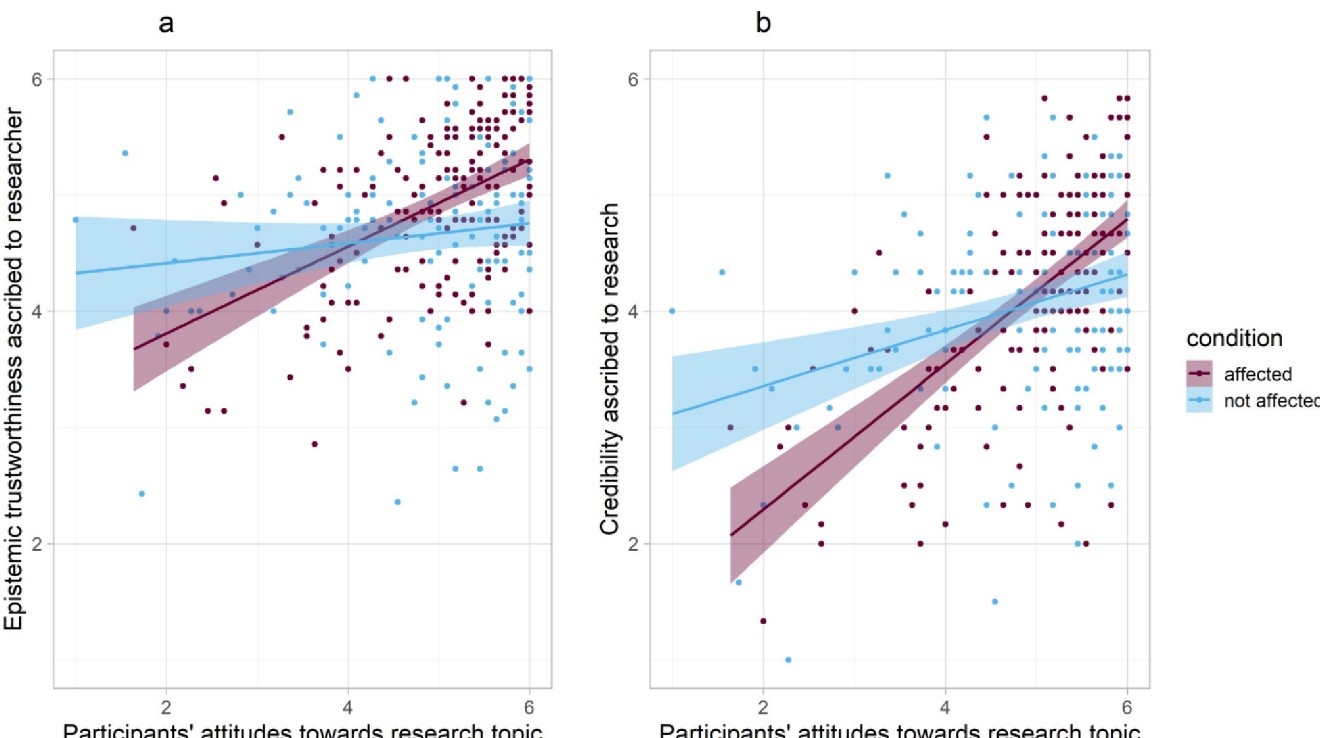

**Fig 1.** Linear regression plots for the interaction effect of attitudes × condition on epistemic trustworthiness (Fig 1A) and credibility (Fig 1B) with 95% confidence intervals: Participants' attitudes towards the research topic moderated how a researcher's disclosure of being personally affected (vs. being not personally affected) by one's own research was perceived.

Lohr's future research findings to be more credible when the researcher was personally affected vs. not affected, *beta* = 0.25, *p* < .001, 95% CI$_{beta}$ [0.12, 0.38]. However, for participants with more negative attitudes (-1 *SD* below sample mean) this effect was significantly reversed: They rated the future research as less credible when the researcher was personally affected vs. not affected, *beta* = -0.18, *p* = .009, 95% CI$_B$ [-0.31, -0.04].

## Discussion

Results from Study 1 suggest that LGBTQ researchers are perceived as more trustworthy and their future findings as more credible when they disclose being personally affected by their research topic (i.e., being queer themselves), but only if perceivers hold positive attitudes towards LGBTQ issues. By contrast, holding less favorable attitudes towards LGBTQ issues lead to more skeptical reactions towards personally affected vs. unaffected researchers. This finding shows that learning about a researcher's personal affection by their research can, indeed, go both ways, as suggested by our theoretical reasoning. On a more general level, our research suggests that public reactions towards "me-search" is a matter of pre-existing attitudes, and, thus, a case of motivated science reception [21, 22].

There are some limitations to this first study: As most people in our sample held rather positive attitudes towards the LGBTQ community (*M* = 4.93, *SD* = 1.02; on a scale from 1 to 6), predicted values on trustworthiness and credibility at the lower end of the attitude spectrum are probably less reliable. Also, we did not control for participants' own identification as belonging to the LGBTQ community. Thus, we cannot differentiate clearly between attitudinal and identity-related effects, which is important because attitudes and identity concerns have a psychologically distinguishable impact on motivated science reception [27, 28]. Additionally, replicating our results in a different domain is necessary to be able to generalize our findings. Another question of generalizability that is left unanswered is how such individual experiences with one personally affected researcher might impact laypeople's perception of the entire field. This calls for more research on the double-edged nature of the moderating effect of preexisting attitudes.

## Study 2

In our preregistered second study (see https://osf.io/c9r4e), we aimed to replicate our findings in a more diverse sample and with a different research topic that has the potential of polarizing participants even more strongly. We used the same design as in Study 1, but changed the proposed research topic to perceptions of vegans and introduced a vegan vs. non-vegan researcher. Again, we hypothesized that laypeople's attitudes towards veganism moderate the effects on trustworthiness as well as credibility of future research. Additionally, we tested whether the effect of one researcher being personally affected by their own research generalizes to the broader perception of their entire field. Furthermore, we also explored whether the moderation by attitudes towards veganism prevailed when controlling for self-identification as being vegan (not included in preregistration).

### Method

**Sample.**  We conducted an a-priori power analysis using G*Power [35] for detecting the hypothesized interaction effect in a moderated multiple regression analysis ($f^2$ = 0.04, based on Study 1, with 1-$\beta$ = 0.90 and $\alpha$ = 0.05, which resulted in a total sample of *N* = 265. Anticipating exclusions (see specified criteria) of comparable size as in the previous study, we aimed for a sample of at least 350 participants.

We collected data from 364 participants via mailing lists and social media. Fifty-seven participants had to be excluded due to our preregistered criteria (see https://osf.io/c9r4e): one participant was younger than 16 years, 31 failed the manipulation check, 10 took less than 20sec viewing the proposal, 12 took less than 3min or more than 20min for participation, 3 had apparently implausible patters of response (i.e., "straight-lining;" identical responses on every single item on more than one questionnaire page in a row). The final sample consisted of $N = 307$ participants (76% female, 23% male, 1 other) who were between 18 and 79 years old ($M = 33.55$, $SD = 13.92$). Approximately half of the sample (50%) was currently studying at a university, further 40% were working and 10% not working, one person was currently in training. Eighty-five percent were currently studying or already held a university degree (social sciences: 49%, humanities: 17%, natural sciences: 14%, life sciences: 8%, engineering: 6% and other 6%). Most participants did not consider themselves as vegans (89%).

**Materials and procedure.** We used the same materials and procedure as in Study 1 (see OSF for full materials: https://osf.io/phfq3/). However, we changed the research topic to "perceptions of vegans". Participants were randomly assigned to two conditions. In the "not personally affected" condition, the researcher Dr. Lohr wrote:

"*I was interested in investigating this research questions not only out of scientific reasons but because, as someone who is not living as a vegan and, thus, not personally affected by my own research, I think we have a need for more evidence-based knowledge regarding the social embedding of vegan lifestyles, which we can acknowledge in everyday life.*"

In the "personally affected" condition, Dr. Lohr wrote:

"*. . . because, as someone who is living as a vegan and, thus, personally affected by my own research, I think we have a need for more evidence-based knowledge regarding the social embedding of vegan lifestyles, which we can acknowledge in everyday life.*"

As dependent variables, we again used the 14-item METI [36] to measure epistemic trustworthiness, but we expanded the measure for credibility of future research by adding one more item ("I would express skepticism towards Dr. Lohr's future findings") to better capture the behavioral aspects of credibility (now: 7 items; Cronbach's α = .86). We also added a measure of participants' evaluation of the entire field (not the specific researcher) as a third dependent variable. This 12-item scale was adapted from a related study [28] (e.g., "I think researchers who do research on that topic sometimes lack competence," "I think it is difficult to apply results from this line of research to reality;" 6-point Likert scale ranging from 1 = "not at all" to 6 = "very much;" Cronbach's α = .85). Next, participants' attitudes towards veganism (i.e., the moderator variable) were measured with a 14-item scale adapted from the attitude measure in Study 1 by changing and adding items (e.g., "I think veganism is exaggerated" (reverse-coded) and "I can imagine being a vegan myself;" 6-point Likert scale ranging from 1 = "not at all" to 6 = "very much;" Cronbach's α = .95).

To reduce exclusions after data collection, participants could proceed only if they answered all attention checks correctly (4 items; multiple choice). We added self-identification as vegan as a control variable ("Do you presently consider yourself a vegan?" yes/no); and an open-ended question about participants' opinion regarding the researcher being personally affected to explore how laypeople rationalize their opinion. These responses were later coded for valence (positive, negative, mixed, or neutral) and content (deductive and inductive coding) by two raters blind to the specific research question (see Appendix C in the supplementary materials, https://osf.io/phfq3/; interrater reliability for valence, Cohen's κ = .86, $p < .01$; and

for content, Cohen's κ = .74, $p < .01$). Again, the questionnaire closed with a sign-up for a lottery and more information as well as a debriefing.

## Results

Our randomized groups did not differ in regard to PES (PES frequency, $p = .147$; PES experiences, $p = .101$). However, they did differ significantly in regard to the general perception of neutrality in science ($p = .049$). Possible implications are addressed in the Discussion. Table 3 summarizes all means, standard deviations, correlations and internal consistencies. In the following, we report our findings for all three dependent variables (trustworthiness, credibility, evaluation of the entire field), consecutively.

**Trustworthiness.** First, we ran the standardized regression model for epistemic trustworthiness. There was neither a significant main effect of condition on epistemic trustworthiness, $beta = 0.04$, $p = .482$, 95% $CI_{beta}$ [-0.07, 0.15] nor a significant main effect of attitudes towards veganism, $beta = 0.07$, $p = .205$, 95% $CI_{beta}$ [-0.04, 0.18]. However, the hypothesized condition × attitudes interaction effect was significant, $beta = 0.22$, $p < .001$, 95% $CI_{beta}$ [0.11, 0.34], increasing the amount of explained variance in trustworthiness by 4% to $R^2_{adj} = .05$. Table 4 summarizes the results. Fig 2A and standardized simple slopes analyses show that participants with more positive attitudes towards veganism (+1 $SD$ above sample mean) trusted Dr. Lohr more when personally affected vs. not affected, $beta = 0.26$, $p = .001$, 95% $CI_{beta}$ [0.11, 0.42]. This conditional effect was reversed for participants with more negative attitudes (-1 $SD$ below sample mean), who trusted Dr. Lohr less when personally affected vs. not affected, $beta = -0.19$, $p = .020$, 95% $CI_{beta}$ [-0.34, -0.03]. The interaction effect remained significant when controlling for participants' self-identification as being vegan ($p < .001$, total $R^2_{adj} = .06$). In secondary analyses, we explored the effects on the two facets of epistemic trustworthiness, separately. The same pattern of interaction effects emerged for both integrity/benevolence ($p < .001$, total $R^2_{adj} = .08$) and expertise ($p = .005$, total $R^2_{adj} = .02$); full analyses are reported in Appendix D in the supplementary materials (see https://osf.io/phfq3/).

**Credibility.** On credibility, there was no significant main effect of condition, $beta = -.07$, $p = .146$, 95% $CI_{beta}$ [-0.17, 0.03] but a significant main effect of attitudes towards veganism, $beta = .35$, $p < .001$, 95% $CI_{beta}$ [0.25, 0.45]. As predicted, the condition × attitudes interaction

**Table 3. Means, standard deviations, correlations, and internal consistencies.**

| Variable | M | SD | α | 1 | 2 | 3 | 4 | 5 | 6 | 7 | 8 |
|---|---|---|---|---|---|---|---|---|---|---|---|
| 1. Expertise | 4.68 | 0.91 | .92 | | | | | | | | |
| 2. Integrity/ benevolence | 4.75 | 0.75 | .90 | .71*** | | | | | | | |
| 3. Epistemic trustworthiness | 4.72 | 0.76 | .94 | .92*** | .93*** | | | | | | |
| 4. Credibility | 3.97 | 0.96 | .86 | .66*** | .70*** | .73*** | | | | | |
| 5. Evaluation of field | 2.95 | 0.78 | .85 | -.40*** | -.43*** | -.45*** | -.64*** | | | | |
| 6. Attitudes towards veganism | 4.26 | 1.23 | .95 | .04 | .15** | .11 | .39*** | -.46*** | | | |
| 7. Neutrality expectation | 4.18 | 0.91 | .86 | .05 | .06 | .05 | .08 | -.26*** | .02 | | |
| 8. PES frequency | 3.30 | 0.68 | .75 | -.13* | -.06 | -.10 | -.03 | -.03 | .11 | .08 | |
| 9. PES experiences | 6.47 | 3.25 | - | -.14* | -.09 | -.12* | -.09 | .03 | .07 | -.02 | .51*** |

*Note.* N = 307.

* indicates $p < .05$.

** indicates $p < .01$.

*** indicates $p < .001$. α represents internal consistencies (Cronbach's α). Variables 1–7 ranged from 1 to 6, variable 8 ranged from 1–5 and variable 9 ranged from 0–15.

**Table 4. Standardized regression results, Study 2.**

| Predictor | beta | beta | $sr^2$ | $sr^2$ |
|---|---|---|---|---|
|  |  | 95% CI |  | 95% CI |
| Epistemic trustworthiness |  |  |  |  |
| condition | 0.04 | [-0.07, 0.15] | .00 | [-.01, .01] |
| attitudes | 0.07 | [-0.04, 0.18] | .01 | [-.01, .02] |
| condition × attitudes | 0.22*** | [0.11, 0.34] | .05 | [.00, .10] |
| Credibility |  |  |  |  |
| condition | -0.07 | [-0.17, 0.03] | .01 | [-.01, .02] |
| attitudes | 0.35*** | [0.25, 0.45] | .12 | [.05, .18] |
| condition × attitudes | 0.25*** | [0.15, 0.35] | .06 | [.01, .11] |
| Critical evaluation of field |  |  |  |  |
| condition | -0.00 | [-0.10, 0.10] | .00 | [-.00, .00] |
| attitudes | -0.41*** | [-0.51, -0.31] | .17 | [.09, .24] |
| condition × attitudes | -0.27*** | [-0.37, -0.18] | .07 | [.02, .12] |

*Note.* N = 307.

*** indicates $p < .001$. Condition is standardized by effect coding (-1 = not personally affected, 1 = personally affected). *beta* represents standardized regression weights. $sr^2$ represents the semi-partial correlation squared.

effect was also significant for credibility, *beta* = 0.25, $p < .001$, 95% CI$_{beta}$ [0.15, 0.35], increasing the amount of explained variance in credibility by 6% to $R^2_{adj}$ = .21. Table 4 summarizes these results. Fig 2B and standardized simple slope analyses qualify the interaction effect: In line with the results for trustworthiness, participants with more positive attitudes (+1 *SD* above sample mean) anticipated Dr. Lohr's future findings to be more credible when personally affected vs not affected, *beta* = 0.18, $p = .016$, 95% CI$_{beta}$ [0.03, 0.32], while the conditional effect for participants with more negative attitudes (-1 *SD* below sample mean) changed its sign, *beta* = -0.32, *SE(B)* = 0.14, $p < .001$, 95% CI$_{beta}$ [-0.47, -0.18]. As before, the interaction effect remained significant when controlling for self-identification as being vegan ($p < .001$, total $R^2_{adj}$ = .21).

**Evaluation of the field.** Third, we investigated whether this moderation effect generalizes to the evaluation of the entire field of veganism research. There was no significant main effect of condition, *beta* = -.00, $p = .989$, 95% CI$_{beta}$ [-0.10, 0.10] but a significant main effect of attitudes, *beta* = -.41, $p < .001$, 95% CI$_{beta}$ [-0.51, -0.31]. Again, we found the hypothesized condition × attitude interaction effect, *beta* = -.27, $p < .001$, 95% CI$_{beta}$ [-0.37, -0.18], increasing the amount of explained variance in critical evaluation by 7% to $R^2_{adj}$ = .27. Again, Table 4 summarizes these results and Fig 2C and standardized simple slopes analyses further qualify the interaction effect: Participants with more positive attitudes towards veganism (+1 *SD* above sample mean) were less critical of research on veganism when Dr. Lohr was personally affected vs. not affected, *beta* = -0.28, $p < .001$, 95% CI$_{beta}$ [-0.41, -0.14]. By contrast, this conditional effect was reversed for participants with more negative attitudes towards veganism (-1 *SD* below sample mean), *beta* = 0.27, $p < .001$, 95% CI$_{beta}$ [0.14, 0.41]. This interaction effect also remained significant when controlling for self-identification as being vegan ($p < .001$, total $R^2_{adj}$ = .28).

**Participants' opinion.** Overall, participants who responded to the open-ended question expressed mostly negative opinions about the researcher being personally affected by his own research (negative: 48%, neutral: 21%, positive: 17%, and mixed: 14%). The most frequently mentioned (negative) remark was that a "me-searcher" might be biased towards their research

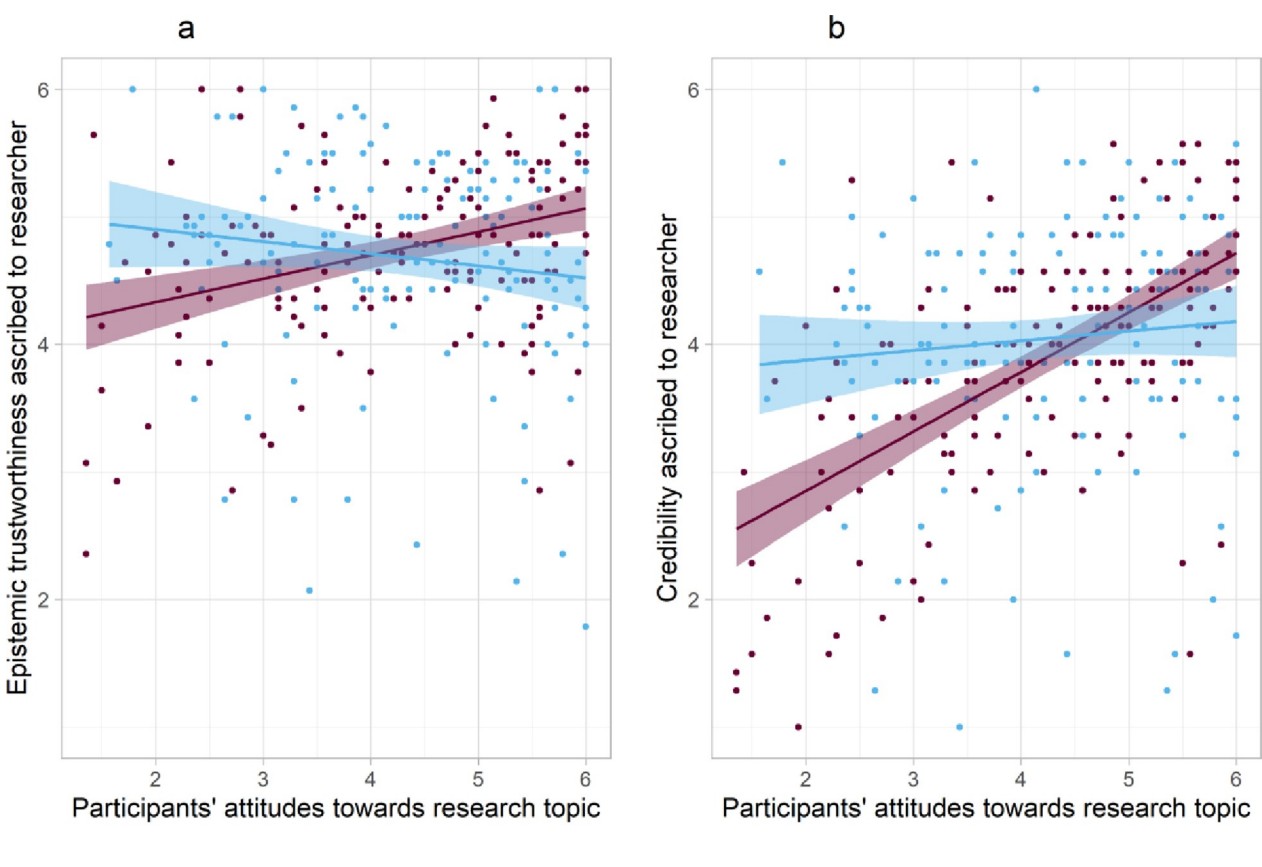

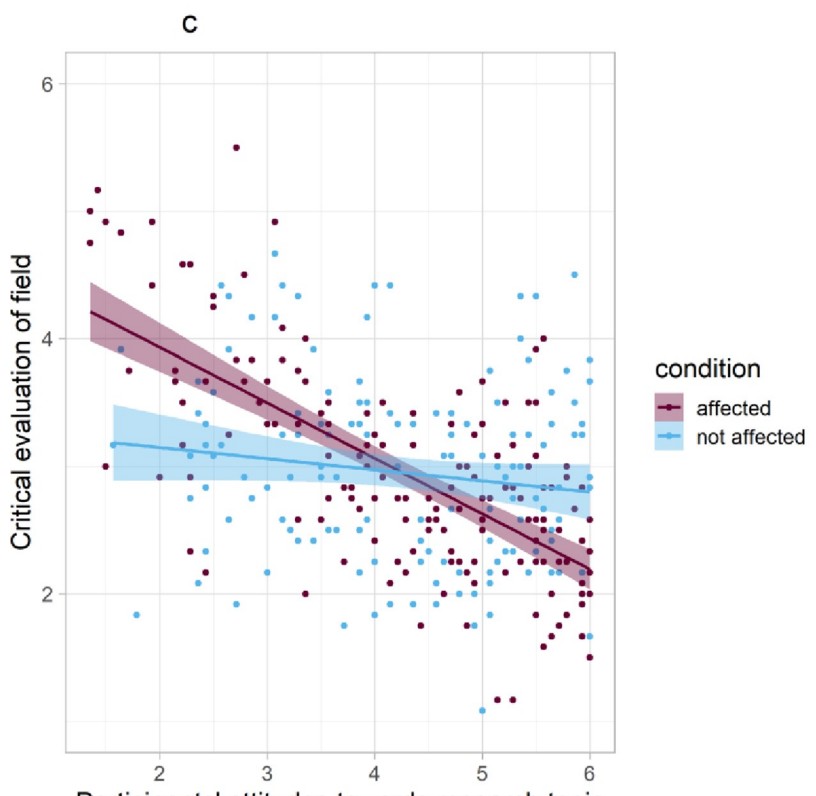

**Fig 2.** Linear regression plots for the interaction effect of attitudes × condition on epistemic trustworthiness (Fig 2A), credibility (Fig 2B) and critical evaluation of the entire field (Fig 2C) with 95% confidence intervals: Participants' attitudes towards the research topic moderated how a researcher's disclosure of being personally affected (vs. being not personally affected) by one's own research was perceived.

(60%; e.g., "*By introducing himself as being affected, I fear he cannot evaluate the results of his research objectively*"). The second most frequently mentioned remark was that such idiosyncratic relevance is irrelevant (24%; e.g., "*It wouldn't make a difference*"). Positive remarks were mentioned less frequently: Participants ascribed more motivation (11%; e.g. "*I think interest, also personal interest, is an important prerequisite for determined research*") or knowledge about the topic (8%; e.g. "*Very good, most likely, he thus is knowledgeable about the subject matter and can conduct the study in a more purposeful manner*") to the "me-searcher", or recognized the transparency (7%; e.g., "*The main thing is transparency. People are always biased, perhaps even unconsciously*"); for more details, see Appendix C in the supplementary materials: https://osf.io/phfq3/).

## Discussion

In Study 2, we replicated the moderation effect of preexisting attitudes on the effect of a researcher disclosing being personally affected (vs. not affected) by their own research on participants' epistemic trustworthiness and credibility ascriptions regarding the research and researcher's future findings. Further, we showed that this effect generalizes to the evaluation of the entire research area. Here, positive attitudes towards veganism determined how learning about an openly vegan researcher impacted participants' perceptions of trustworthiness and credibility as well as the evaluation of the entire field of veganism research compared to learning about a non-vegan (i.e., non-affected) researcher. Participants who held more positive attitudes towards veganism reported more trust, higher anticipated credibility of future findings, and a less critical evaluation of the field when confronted with a vegan researcher. Conversely, for participants with less positive attitudes this effect was reversed. The moderation by positive attitudes towards veganism persisted when controlling for participants' self-identification as vegans. Overall, the interaction effects observed in Study 2 explained similar amounts of variance as in Study 1 (epistemic trustworthiness: 3% vs. 4%, and credibility: 4% vs. 6%). Further, qualitative analyses revealed that most participants reported negative–or, at least, mixed–perceptions of a "me-searcher" (e.g., "me-searchers" may be biased, but also highly motivation and knowledgeable), which corroborated our theoretical prediction that "me-search" may be a double-edged sword. Interestingly, these qualitative findings seem somewhat contradictory to the quantitative findings, according to which there was no main effect of researchers' idiosyncratic affection by their research topic.

In Study 2, one caveat is that the groups differed significantly in regard to participants' general expectations of neutrality in science. Participants who read about the personally affected researcher had weaker expectations of neutrality; yet, when added to the regression model as a control, the pattern of results remained unchanged (see Appendix E in the supplementary materials, https://osf.io/phfq3/). Further, as a second caveat, we show that participants generalized their perceptions to the overall field of veganism research. However, this research area might be considered quite narrow and, thus, future research should investigate how far such generalization processes stretch out to perceptions of broader areas of research (e.g., health psychology).

## General discussion

In two studies, we show that laypeople's perception of researchers who disclose being personally affected by their own research can be positive as well as negative: The effect of such "me-

search" was moderated by laypeople's preexisting attitudes. Queer or vegan researchers were perceived as more trustworthy and their future findings were anticipated to be more credible when participants had positive, sympathizing attitudes towards the related research object (i.e., LGBTQ community or veganism). When participants' attitudes were less positive, this pattern reversed. In Study 2, we extended our research from individualized perceptions of single researchers and their findings to evaluations of the entire field of research. Participants who were confronted with a personally affected researcher seemed to consider this person a representative example and generalized their judgment to their evaluation of the entire (though here quite narrow) research area.

We explored epistemic trustworthiness in more detail in both studies, namely the cognitive-rational facet of expertise and the affective facet of integrity/benevolence: Both were impacted by researchers' disclosure of being personally affected, although effect sizes for expertise were descriptively smaller than for integrity/benevolence. This points to "me-search"– when received positively–possibly adding to the perception of competence-related aspects like a deeper knowledge of a phenomenon (e.g., via anecdotal insights) [12–14] and, even more so, warmth-related aspects like seeming more sincere, benevolent, transparent and, thus, approachable [15, 16, 41]. Disclosing such personal interest in a scientific endeavor might be able to bridge the stereotypical perception of cold and distant "science nerds" by revealing passionate, human and, thus, more relatable side of a researcher. When received negatively, however, "me-search" might be regarded as harboring vested interests, which casts doubts on a researcher's neutrality and objectivity [8–11, 42].

In general, the main models tested here explained between 5% and 28% of variance which may not appear impressive at first glance. However, our studies posed a very strict test of the effects of "me-search" by only using a subtle manipulation sparse in information followed by measures of very specific perceptions which might have contributed to an understatement of the real-world impact.

"Me-search" neither automatically sparks trust nor mistrust in laypeople, even if their explicit opinions seem rather negative. In line with assumptions from motivated science reception [22, 43], our findings suggests that the ambivalence of the fact that a researcher is personally affected can be seized as an opportunity to interpret the situation in a manner that best fits to preexisting attitudes: Researchers, their findings and even their entire field of research are evaluated–even before learning about specific findings–based on prior attitudes towards the research topic. We show in Study 2 that the moderation effect of participants' positive attitudes towards the respective research topic (i.e., veganism) prevails when controlling for self-identification with the topic (i.e., being a vegan). This suggests that, indeed, in motivated reasoning attitudinal and identity-related processes can be differentiated: Here, social identity protection could be ruled out as alternative explanation for the effects of pre-existing attitudes. Noteworthily, we demonstrate that motivated science reception already operates when the results are not (yet) known. This points towards a perceptual filter made up of pre-existing attitudes that is activated when confronted with scientific information and leads to biased pre-judgments: Ambivalent cues (i.e., "me-search") are prematurely interpreted in line with prior attitudes without actually knowing whether the new scientific information will be attitude-consistent or inconsistent (when, later, results are reported).

## Future research

Future research on the motivated reception of "me-search" should focus on three open questions. First, while we consider it a strength of our studies that the results of the proposed research project were not yet known, it might be interesting to see how being personally

affected or not interacts with the perceived direction of the communicated scientific results (e.g. supporting vs. opposing a certain position): To what extent can the first, premature evaluation of a "me-searching" researcher be adapted if the actual results are inconsistent with this pre-judgment?

Second, the investigation of what specific characteristics of "me-search" are instrumentalized by benevolent or skeptical perceivers might not only provide practical tips on how to handle being personally affected (e.g., in science communication) but also important theoretical insights on the building blocks of trust in science and researchers (see discussion above regarding the effects on the facets of epistemic trustworthiness). As one example, knowing that a qualitative level of knowledge is highly valued could further research on the trust-benefit of enriching statistical evidence with anecdotal and narrative elements [44, 45]. As second example, we argue that researchers' self-disclosure of being personally affected by their research might signal transparency and, thus, improve the perception of the trust facets integrity and benevolence. Yet, even the disclosure of *not* being personally affected could have such an effect on a researcher's reputation and, at the same time, it might be less ecologically valid (as, presumably, it is rather unusual to explicitly state to *not* be affected by something). Introducing a control group without any information about a researcher's relation towards their research object might bring light to this.

Third, we demonstrated the moderation effect of preexisting attitudes for two research areas (i.e., LGBTQ and veganism) and in different populations. Yet, further research should investigate whether this effect will hold up for other areas, more diverse samples and different kinds of "me-search", as well. For example, in some research fields being personally affected by the research might be perceived as more morally charged than in others and, thus, having stronger polarizing effects [46]: While, in veganism-research, "me-search" might be grounded in an ideological choice (e.g., thinking its morally wrong to consume animal products and, thus, being vegan), having a stroke and, following, studying stroke-related brain plasticity is likely perceived as less ideological. Also, different scientific methods (typically) used in a field might impact the perceptions of "me-search" depending on how prone for subjectivity these methods are perceived to be (e.g., qualitative "me-search" like autoethnographic analyses might be perceived more critically than when using seemingly objective, quantitative methods like physiological measures). Further, researchers who are not *directly* personally affected by their research but "merely" interested in something for personal reasons (e.g., being highly empathetic towards queer concerns without identifying as queer) might not profit from disclosure of such personal motivations: Such researchers might be perceived as impostors [47] lacking the expertise stemming from directly firsthand experiences.

## Practical implications

Finally, for the applied perspective on public engagement with science, it should again be noted, that motivated reasoning processes are activated even before specific results are presented (e.g. before hearing a talk or reading about a study). This might be important, as judgments are quickly formed and remembered [48, 49] and, therefore, the first impression of a researcher might set the tone for further interactions and, particularly, for the acceptance and implementation of their findings. This emphasizes the importance of researchers knowing their audience (and their attitudes) when engaging in science communication.

Of course, there are also ethical considerations concerning "me-search": Researchers should always declare any conflict of interests when conducting research [50, 51]. Failing to disclose being personally affected by one's own research might backfire severely on researchers' reputation–especially concerning their trustworthiness and the credibility of their findings–and in

particular, when this information is disclosed by someone else and not themselves. At least for achieving positive reputational effects, it seems researchers need to freely initiate the disclosure of limitations and problems themselves [41, 52]. A possible solution for reaping all the benefits and protecting against the potential harms of engaging in "me-search" might be to actively seek out mixed research teams. Including affected as well as non-affected individuals in research projects might be worth considering from the stance of the public's trust in science: It enables deep, even personal insights to the studied phenomenon, while still securing balanced perspectives and impartiality.

## Conclusion

Neuroanatomist Jill Bolte Taylor became famous for turning her "stroke of fate" into productive and well-selling "me-search". Yet, she was praised as well as heavily criticized for mixing her personal and scientific motivations: When research is also "me-search", it can be perceived positively as well as negatively depending on laypeople's preexisting attitudes towards the research object. Researchers who disclose being personally affected by their own research can benefit from this disclosure in terms of trustworthiness and credibility when it is perceived by laypeople with positive attitudes; however, for audiences with more negative attitudes this effect is reversed and disclosure can be harmful. One experience with a personally affected researcher might be enough to impact the evaluation of the whole field. Thus, openly acknowledging "me-search" in one's research is an ambivalent matter and its communicative framing as well as the targeted audience should be well considered.

## Author Contributions

**Conceptualization:** Marlene Sophie Altenmüller, Leonie Lucia Lange, Mario Gollwitzer.

**Data curation:** Marlene Sophie Altenmüller, Leonie Lucia Lange.

**Formal analysis:** Marlene Sophie Altenmüller, Leonie Lucia Lange.

**Investigation:** Marlene Sophie Altenmüller, Leonie Lucia Lange.

**Methodology:** Marlene Sophie Altenmüller, Leonie Lucia Lange.

**Project administration:** Marlene Sophie Altenmüller.

**Resources:** Marlene Sophie Altenmüller, Leonie Lucia Lange.

**Supervision:** Mario Gollwitzer.

**Validation:** Marlene Sophie Altenmüller.

**Visualization:** Marlene Sophie Altenmüller.

**Writing – original draft:** Marlene Sophie Altenmüller.

**Writing – review & editing:** Marlene Sophie Altenmüller, Mario Gollwitzer.

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
