## [Decision Letter · Decision Letter 0]

19 Mar 2021

PONE-D-20-38206

When research is me-search: Researcher interests affect laypeople’s trust in science depending on their pre-existing attitudes

PLOS ONE

Dear Dr. Altenmüller

Thank you for submitting your manuscript to PLOS ONE. After careful consideration, we feel that it has merit but does not fully meet PLOS ONE’s publication criteria as it currently stands. Therefore, we invite you to submit a revised version of the manuscript that addresses the points raised during the review process.

VI have now received three reviews of your MS. All three reviewers see merit in the research presented in the paper. Two reviewers recommend publication after minor revision, whereas the third has recommended major revision, in particular in relation to the statistical analysis applied.  In view of the comments of reviewer two, I am recommending major revision and would like to invite you to revise and resubmit the MS. As well as addressing al of the reviewers comments, please address the issues associated with the regression analysis raised by reviewer 2 in your resubmission.

I have  received three reviews of your MS. All three reviewers see merit in the research presented in the paper. Two reviewers recommend publication after minor revision, whereas the third has recommended major revision, in particular in relation to the statistical analysis applied.  In view of the comments of reviewer two, I am recommending major revision and would like to invite you to revise and resubmit the MS. As well as addressing al of the reviewers comments, please address the issues associated with the regression analysis raised by reviewer 2 in your resubmission.

Please submit your revised manuscript by 1st June 2021 If you will need more time than this to complete your revisions, please reply to this message or contact the journal office at plosone@plos.org. Please include the following items when submitting your revised manuscript:

We look forward to receiving your revised manuscript.

Kind regards,

Lynn Jayne Frewer, MSc PhD

Academic Editor

PLOS ONE

Journal Requirements:

Reviewers' comments:

Reviewer's Responses to Questions

**Comments to the Author**

1. Is the manuscript technically sound, and do the data support the conclusions?

Reviewer #1: Partly

Reviewer #2: No

Reviewer #3: Yes

2. Has the statistical analysis been performed appropriately and rigorously? 

Reviewer #1: Yes

Reviewer #2: Yes

Reviewer #3: Yes

3. Have the authors made all data underlying the findings in their manuscript fully available?

Reviewer #1: Yes

Reviewer #2: Yes

Reviewer #3: Yes

4. Is the manuscript presented in an intelligible fashion and written in standard English?

Reviewer #1: Yes

Reviewer #2: Yes

Reviewer #3: Yes

5. Review Comments to the Author

Reviewer #1: I like this line of research very much and think it's useful and largely well done. I wish you had a more diverse and larger sample but ... maybe next time. It's not necessary, it'd just make the results (especially in study 1 where you don't have a lot of non-LGBTQ friendly respondents.

Some concerns I have ...

1. Ecological validity for the 'non-affected' condition: It's not clear to me why anyone would ever say that they're studying something because they don't identify with it. I think, more likely, someone might just NOT say why they're studying something and just describe what they're studying with no context. Alternatively, a scientist might say they study something because (a) it personal affects them, or (b) it's just an interesting set of puzzles, or (c) both.

2. It's also probably worth making clear that disclosing that research is personally affecting the respondent is conceptually different from saying that a respondent is motivated by benevolence. In this regard, I worry a bit that both respondents are kind of making a (weak) benevolence claim that might be attenuating the results. As future research, you might consider research that adds a clear "and my motivation is to help this community ..." message as a condition. I also don't really understand the last sentence in either condition. They seem fairly vague.

3. Is 'affected' the right word? 'Interested' (as in conflict of interest) seems closer but that word may have too much baggage. But 'affected' has baggage too and we're all 'affected' by research in these areas. Maybe 'involved'?

4. I do not understand your argument about how the second study rules out social identity protection. More generally, I don't know that social identity protection and attitude protection are necessarily incompatible given that my identity might make it more likely that I hold certain evaluative beliefs (aka the basis of attitudes). And the fact you're vegan doesn't mean that veganism is a core part how you identify. If you want to get at identity protection, I'd want more evidence than this. I think you'd be better just to talk about motivated reasoning in a more general sense.

Some technical suggestions ...

1. You sometimes present percentages with two decimals, which is essentially four decimals (inasmuch as 21.34% = .2134), but your sample is less than a 1K people. This would seem to suggest that you're trying to be more precise than is reasonable. Id' just stick to two decimals throughout unless you have a good reason.

2. Does it make sense to call your behavioral trust measure 'credibility'? I worry a bit about this because there's so much variability in the trust literature when it comes to what we call the various constructs. In this regard, I note that credibility is often conceptualized as competence (even going back to Hovland in 1951). I think it might be safer just to talk about 'behavioral trust as willingness' to be vulnerable as that fits with the Roger/Davis/Schoorman model that the Muenster epistemic inventory is derived.

3. It'd be great if your tables and figures were a bit more descriptive such that they can 'stand alone'. In this regard, you might add sample sizes as well as variable range. The figure, in particular, are impossible to understand without referring to the text. In those cases, a detailed note is one important step but it'd also help to better label your axes and use a readable font.

4. Does it really make sense in the second study to include the bit about the overall field? In this regard, is vegan research really 'the field'? Why wouldn't the field be vegetarian studies? Or food studies? Or nutrition? Or biological science? You define field so narrowly that I'm not sure what this additional element adds.

5. You note that the effect sizes for expertise were smaller than the effect sizes for integrity/benevolence but can you really say that? I don't recall you testing the size of the difference, noting that the estimates have error associated with them such that they could easily overlap.

Reviewer #2: Thank you for making your data and analysis scripts openly available. The code seems to run fine and I can reproduce the results.

Method. Sample size

1. I appreciate that you took the effort to provide a power analysis, but post-hoc power is not particularly informative. What the reader needs to know is: given a specific sample size (N = 314), what is the smallest effect one can detect? In this case, it is r = .16 80% of the time, or r = .20 95% of the time. Or, since you’re using means comparisons, this is roughly a d = .32 80% of the time, or a d = .41 95% of the time. [I used the pwr() package for this].

I recommend adjusting this description to provide this information.

Results

What I would like to see presented are the partial r correlations from the regression. It seems that, from your description, the most informative predictor here is ‘participant positive attitude’ and that needs to be compared to the other IV. Yes, there is an interaction, but the slopes are all positive in figure 1a and 1b, suggesting it’s simply a difference in strength rather than a manipulation of the direction of the effect. This can be seen in the figures (which are nice by the way, but please upload higher quality versions).

I recommend reporting the full regression model in a table along with partial correlations and standardised coefficients rather than only the raw coefficients in text.’

It would really be nice to have all the results in one table so that they can be easily compared across studies: I suggest including standardized betas and partial correlations.

‘Evaluation of the field’. It’s claimed that participants positive attitudes towards veganism moderate evaluations such that the effect changes direction, and the text points to Figure 2c. I think this must be a mistake: the slopes and CIs reported in-text are simply opposite of each other (eg b = -.43 and b = .43). Further, the graphs 2c depicts only negative slopes.

Figure 2a, in contrast, shows slopes with opposing directions. Please double check this.

The adjusted R squared for trustworthiness is tiny: R = .05. I’m not sure what to make of these effects, leaving me wondering about the value of this model. The size is similar between both studies, suggesting this is probably accurate.

Again, without partial R values, it’s hard for the reader to compare the relative strength of these effects and judge their merit.

The adjusted R-squared values for credibility and evaluation of the field are upwards of R = .20, which is more convincing (and this is noted in the discussion).

In the Study 2 discussion, the authors suggest that “The moderation…persisted when controlling for self-identification as a vegan… this has to be interpreted carefully.” I suggest you remove this caution or ignore the effect: either interpret something or don’t. There is no carefulness when you report an effect with an accompanying p-value. If you are not convinced that this result is not a fluke then simply describe your data. Either way, please remove this claim of caution.

Finally, you note that the expectations of neutrality was affected by the manipulation and claim this effect is “small” (d = .22). This is actually more of an average effect size in social/personality studies (see Funder & Ozer, 2019 and Gignac & Szodorai, 2016). I wouldn’t be so quick to dismiss it.

Additionally, earlier on in the manuscript (under Main effect of being affected) you interpret a similarly sized d = .25 at face value. If you have determined an alpha level of .05 as your criteria (which you did, and you also can see that smallest effect size based on the power analysis), you should stick to that criteria throughout and simply interpret the effects.

Finally, I admit that I am a bit puzzled by the overall pattern here: participants were less critical when Dr Lohr was affected; they saw results as more credible when Dr Lohr was affected; they trusted the results more when Dr Lohr was demonstrably biased?

What would happen if you asked the participant to rate whether the researcher is biased?

General Discussion

This is again an issue in the general discussion: you claim that ‘when participant attitudes were less positive, this pattern flipped…” but this pattern did not flip. We simply see a reduction in strength.

Conclusions

In sum, I think the key variable here is participant attitudes towards the topic. I’m not convinced of the moderation effect here for several reasons.

First, it does not reverse the direction of the effect: this is key for the claim

Second, the moderation itself is small. There’s not much of a difference between any two given slopes (with the exception of Figur 2a).

Third, the whole idea strikes me as strange: participant attitudes drive the effect always in a positive direction. Even when the researcher admits to being biased, participants don’t evaluate him more negatively. Plus, ‘bias’ itself wasn’t measures as far as I saw.

I think this is an interesting set of studies but it needs a bit more data to rule out some possible effects and I think the interpretation needs to be revised. I’d like to see these points addressed in a revision and I hope my comments are helpful in that regard.

References

Funder, D. C., & Ozer, D. J. (2019). Evaluating effect size in psychological research: Sense and nonsense. Advances in Methods and Practices in Psychological Science, 2(2), 156-168.

Gignac, G. E., & Szodorai, E. T. (2016). Effect size guidelines for individual differences researchers. Personality and individual differences, 102, 74-78.

Reviewer #3: This paper examines how peoples’ attitudes toward a topic influence their evaluation of research and researchers’ trustworthiness - in the context of research topics which have personal relevance to the researchers.

I believe the aim of this study is very interesting and timely, and a great addition to the research; the study is well done. Also positive is the osf documentation and preregistration of study 2.

While I am quite enthusiastic about the paper, I also have a few issues to remark:

Abstract:

Reading your abstract for the first time, I was a bit confused whose attitudes and interests you were talking about when and what you were measuring – Maybe lead the reader straighter to the point, instead of mirroring your introduction (even though the intro read quite smoothly)?

Methods and Results:

• Please report the results of the factor analyses justifying the scales for testing your hypotheses, can be in the supplement as well.

• p values should be reported in accord with APA reporting guidelines, not p < .05.

• Clearly state what you are testing when comparing the groups with attitudes +- 1SD from mean. “A regression was performed…” or as appropriate.

• Table 1,2 and probably a question of taste: I don’t know of a convention in which Cronbach’s alphas are displayed in the diagonal of a correlation table. Since they’re conceptually something different than correlations, I would suggest moving them to a single column.

• I don’t see a theoretical reason why you should test trustworthiness, and then the two scales separately, would you please justify this? (I think this should be avoided, if there is no indication from theory, and since the results are interdependent anyway. Reminds me of MANOVA)

Discussion (and maybe Introduction):

Finally, and probably most importantly, I am a bit unsure about your interpretation of the main effect on trustworthiness following the experimental variation – you discuss two ideas, and only in the “future research” section: a) there is a preference for anecdotal evidence vs. other types of evidence, i.e. someone affected has some type of special access to a research topic, and b) disclosing being affected / transparency signals benevolence (by the way, in line with intentionalist models of communication). Don’t your results – when dividing the trustworthiness measure into expertise and benevolence/integrity provide some (sure, provisional) evidence that allow you to dive deeper into these explanations in your argumentation? I believe your explanations of this effect could be argued more thoroughly. To that point, while I think it is alright that you only pose an exploratory research question in the introduction, here, you actually almost only provide anecdotal evidence yourselves to argue that question (which you may want to be clear about).

6. PLOS authors have the option to publish the peer review history of their article (what does this mean?). If published, this will include your full peer review and any attached files.

Reviewer #1: No

Reviewer #2: No

Reviewer #3: No

---

## [Author Response · Author response to Decision Letter 0]

7 Jun 2021

Response to Reviewers

First, we would like to thank all reviewers for raising valid concerns regarding the first version of our manuscript.

Reviewer #1: 

I like this line of research very much and think it's useful and largely well done. I wish you had a more diverse and larger sample but ... maybe next time. It's not necessary, it'd just make the results (especially in study 1 where you don't have a lot of non-LGBTQ friendly respondents.

Thank you. We have included this aspect as a potential limitation of our research and an outlook to future research to the General Discussion, p. 28: “Yet, further research should investigate whether this effect will hold up for other areas, more diverse samples and different kinds of “me-search”, as well.”

Some concerns I have ...

1. Ecological validity for the 'non-affected' condition: It's not clear to me why anyone would ever say that they're studying something because they don't identify with it. I think, more likely, someone might just NOT say why they're studying something and just describe what they're studying with no context. Alternatively, a scientist might say they study something because (a) it personal affects them, or (b) it's just an interesting set of puzzles, or (c) both.

Thank you for this comment. First, based on your feedback, we have now refined what we mean by “me-search” and what - in our view - is the opposite of “me-search” (see p. 3). We now clarify that “me-search” or “being affected by a certain research topic” means pursuing a scientific question when the answer to that question is idiosyncratically relevant for the individual researcher as opposed to when the answer is relevant per se. 

Regarding the design of our studies: Our intention in construing the “not affected” condition in that manner was to keep the factor “self-disclosure” constant. This means that in both conditions the research discloses his personal relation to the topic which is either idiosyncratically relevant (as the researcher is directly affected by it) or not idiosyncratically relevant (as the researcher is not directly affected by it). However, we agree that this way we might have involuntarily reduced the ecological validity as a byproduct. We now voice this concern in the General Discussion (p. 27-28: “Yet, even the disclosure of not being affected could have such an effect on a researcher’s reputation and, at the same time, it might be less ecologically valid (as, presumably, it is rather unusual to explicitly state to not be affected by something).”

2. It's also probably worth making clear that disclosing that research is personally affecting the respondent is conceptually different from saying that a respondent is motivated by benevolence. In this regard, I worry a bit that both respondents are kind of making a (weak) benevolence claim that might be attenuating the results. As future research, you might consider research that adds a clear "and my motivation is to help this community ..." message as a condition. I also don't really understand the last sentence in either condition. They seem fairly vague.

This is an interesting point, thanks! With our studies, we are trying to investigate whether respondents actually make these benevolence inferences from researchers’ disclosure of “me-search”, and our results suggest that they do – if they hold a generally positive attitude towards the respective research topic. 

Regarding the phrasing of our manipulation, our intention was to keep the research goal (more evidence-based knowledge) constant while varying the driving interest (affected vs. not affected), without producing too much demand by a public statement about desired outcomes (e.g., to support the community’s demands). We agree that both conditions might signal some benevolence (we argue via transparency) and that it would be interesting to test such a benevolence inference process more directly. Thus, in the General Discussion, we suggest to test this in future research thoroughly by introducing a more neutral control group: “Introducing a control group without any information about a researcher’s relation towards their research object might bring light to this.” (p. 28).

3. Is 'affected' the right word? 'Interested' (as in conflict of interest) seems closer but that word may have too much baggage. But 'affected' has baggage too and we're all 'affected' by research in these areas. Maybe 'involved'?

With this concern you hit the mark on something we also debated at length before deciding on the word “affected”. We feel, the German word „betroffen sein“ is semantically a combination of “being affected by something” and “being involved in something”. We agree that “involved” might have “less baggage” but it does not reflect the directly personal aspect of me-search (i.e., idiosyncratic relevance). For example, you could already be “involved” in something when a team member is affected. Further, “interested” can also be very unpersonal/neutral (e.g., when I find something scientifically interesting). Thus, we decided on “being affected” to best capture the German “betroffen sein”. We added more clarification in the Introduction (p.3) on what we exactly mean by “me-search”, as mentioned in an Author Response above, by adding the term “idiosyncratic relevance”. Still, we use “being affected” to reflect that it is not just a personal relevance (e.g., when debating whether to become vegan or strongly emphasizing with a certain group) but that you are already affected by the situation (e.g., when you already are vegan). We also extended our discussion on different kinds of me-search in the General Discussion section in line with this reasoning: “Further, researchers who are not directly personally affected by their research but “merely” interested in something for personal reasons (e.g., being highly empathetic towards queer concerns without identifying as queer) might not profit from disclosure of such personal motivations: Such researchers might be perceived as impostors [46] lacking the expertise stemming from directly firsthand experiences.” (p.28).

4. I do not understand your argument about how the second study rules out social identity protection. More generally, I don't know that social identity protection and attitude protection are necessarily incompatible given that my identity might make it more likely that I hold certain evaluative beliefs (aka the basis of attitudes). And the fact you're vegan doesn't mean that veganism is a core part how you identify. If you want to get at identity protection, I'd want more evidence than this. I think you'd be better just to talk about motivated reasoning in a more general sense.

We fully agree that identity protection and attitude protection might go hand in hand (as you suggest, a social identity makes it most certainly more likely to have some corresponding attitudes). However, while these two might often co-occur, they still could be grounded in different psychological processes. In Study 2, we find some evidence for this.

Here, we asked participants specifically whether they “identified” as vegan (however, we did not ask for the strength or centrality of this identity). This identification as vegan did not have any independent effects and did not diminish the other effects when added as control variable in our regression models. At first, glance this seems at odds with prior findings on motivated reception being also driven by identity protection efforts (e.g., see studies by Nauroth and colleagues: https://onlinelibrary.wiley.com/doi/abs/10.1002/ejsp.1998, https://journals.plos.org/plosone/article?id=10.1371/journal.pone.0117476, https://doi.org/10.1177/0963662516631289). This might be due to the specific situation we presented: In contrast to most research on motivated reception, we did not confront participants with specific findings or (pro/con) statements but an open-ended situation where results were not yet known. However, while identity did not play a role here, we did find an effect of attitudes in line with motivated reception. Thus, we interpret this as indication that this motivated reception effect based on participants’ attitudes might be functionally different from motivated reception effects based on identity protection. We argue that attitudes might work as broader perception filter that functions even before specific content (i.e., the scientific results) is known, while identity protection might need a specific threat (e.g., scientific evidence calling into question beliefs central to an identity) to lead to motivated responses (e.g. discrediting researchers and their findings when knowing what they say). 

As these conclusions are, indeed, tentative, we refrain from overemphasizing their implications by only shortly mentioning them in the General Discussion (see p. 27-28: “This suggests that, indeed, in motivated reasoning attitudinal and identity-related processes can be differentiated: Here, social identity protection could be ruled out as alternative explanation for the effects of pre-existing attitudes.”).

Some technical suggestions ...

1. You sometimes present percentages with two decimals, which is essentially four decimals (inasmuch as 21.34% = .2134), but your sample is less than a 1K people. This would seem to suggest that you're trying to be more precise than is reasonable. Id' just stick to two decimals throughout unless you have a good reason.

Thank you for the suggestion, we changed the display of our percentages accordingly.

2. Does it make sense to call your behavioral trust measure 'credibility'? I worry a bit about this because there's so much variability in the trust literature when it comes to what we call the various constructs. In this regard, I note that credibility is often conceptualized as competence (even going back to Hovland in 1951). I think it might be safer just to talk about 'behavioral trust as willingness' to be vulnerable as that fits with the Roger/Davis/Schoorman model that the Muenster epistemic inventory is derived.

We fully agree, that trust-related concepts are defined differently by different people. Thus, we totally see the point in clearly defining the terms as we used them in our own research: We use the term “credibility” in clear (verbal) contrast to the term “trustworthiness” to make a distinction between person-related and evidence-related trust/credibility. Our factor analyses support this differentiation empirically (see Appendix A https://osf.io/phfq3/?view_only=a3694575674944fababa32b696e6e645). In addition, we added a clarifying note in the manuscript: p. 7 “[…] Of note, we use the term “credibility” to differentiate evidence-related trust/credibility from person-related trust/credibility, i.e. “trustworthiness”.” 

3. It'd be great if your tables and figures were a bit more descriptive such that they can 'stand alone'. In this regard, you might add sample sizes as well as variable range. The figure, in particular, are impossible to understand without referring to the text. In those cases, a detailed note is one important step but it'd also help to better label your axes and use a readable font.

We added more information to our tables and figures. The figures were uploaded in a higher quality and with better readable labels.

4. Does it really make sense in the second study to include the bit about the overall field? In this regard, is vegan research really 'the field'? Why wouldn't the field be vegetarian studies? Or food studies? Or nutrition? Or biological science? You define field so narrowly that I'm not sure what this additional element adds.

We agree, “veganism research” can only be considered as a very narrow field of research. It might be interesting to investigate how far the generalization stretches out to broader fields as you suggest. The point we make here might be considered a first step, showing that individual experiences of “me-search” are generalized to a broader circle of science reception. In Study 2, we discuss this now as a caveat (p.25: “Further, as a second caveat, we show that participants generalized their perceptions to the overall field of veganism research. However, this research area might be considered quite narrow and, thus, future research should investigate how far such generalization processes stretch out to perceptions of broader areas of research (e.g., health psychology).” And we added a small note in our General Discussion (p. 25: “Participants who were confronted with a personally affected researcher seemed to consider this person a representative example and generalized their judgment to their evaluation of the entire (though here quite narrow) research area.”).

5. You note that the effect sizes for expertise were smaller than the effect sizes for integrity/benevolence but can you really say that? I don't recall you testing the size of the difference, noting that the estimates have error associated with them such that they could easily overlap.

Indeed, we only compared these descriptively. We make this clearer now, by adding “descriptively” in the respective sentence (p. 26 “[…], although effect sizes for expertise were descriptively smaller than for integrity/benevolence.”).

Reviewer #2: 

Thank you for making your data and analysis scripts openly available. The code seems to run fine and I can reproduce the results.

Method. Sample size

1. I appreciate that you took the effort to provide a power analysis, but post-hoc power is not particularly informative. What the reader needs to know is: given a specific sample size (N = 314), what is the smallest effect one can detect? In this case, it is r = .16 80% of the time, or r = .20 95% of the time. Or, since you’re using means comparisons, this is roughly a d = .32 80% of the time, or a d = .41 95% of the time. [I used the pwr() package for this].

I recommend adjusting this description to provide this information.

Thank you for the recommendation. We changed our post hoc power analyses in Study 1 to sensitivity analyses as you suggested. We used G*Power again and arrive at the same results as you using the pwr() package (p.8: “We conducted sensitivity analyses using G*Power [34] for determining which effect sizes can detected with this sample in a moderated (multiple) regression analysis: At α=0.05 and with a power of 80%, small-to-medium effects (f²≥0.03) can be detected with this sample”).

Results

What I would like to see presented are the partial r correlations from the regression. It seems that, from your description, the most informative predictor here is ‘participant positive attitude’ and that needs to be compared to the other IV. Yes, there is an interaction, but the slopes are all positive in figure 1a and 1b, suggesting it’s simply a difference in strength rather than a manipulation of the direction of the effect. This can be seen in the figures (which are nice by the way, but please upload higher quality versions).

I recommend reporting the full regression model in a table along with partial correlations and standardised coefficients rather than only the raw coefficients in text.’

It would really be nice to have all the results in one table so that they can be easily compared across studies: I suggest including standardized betas and partial correlations.

Thank you for the suggestions. We now report standardized regression analyses and included tabular displays of the standardized regression coefficients and semi-partial correlations for all our multiple regression analyses (see Table 2 and 4). Further, we uploaded the figures in better quality and better readable labels.

‘Evaluation of the field’. It’s claimed that participants positive attitudes towards veganism moderate evaluations such that the effect changes direction, and the text points to Figure 2c. I think this must be a mistake: the slopes and CIs reported in-text are simply opposite of each other (eg b = -.43 and b = .43). Further, the graphs 2c depicts only negative slopes.

Figure 2a, in contrast, shows slopes with opposing directions. Please double check this.

Here, we are talking about the simple slopes which are reversed (slope of the predictor when the moderator is held at +/-1 SD). These simple slopes analyses are not directly reflected in the slopes of our figures as we here display the pattern from a different “angle” to help illustrate the conditional effects (mean differences between conditions at different values of the moderator): Here, the moderator is printed on the x-axis (instead of being displayed as slopes held at +/- 1 SD) and the slopes are separated by condition (instead of being displayed on the x-axis). To prevent confusion, we now clarified that in our phrasing in both Results sections (see also below).

The adjusted R squared for trustworthiness is tiny: R = .05. I’m not sure what to make of these effects, leaving me wondering about the value of this model. The size is similar between both studies, suggesting this is probably accurate.

Again, without partial R values, it’s hard for the reader to compare the relative strength of these effects and judge their merit.

The adjusted R-squared values for credibility and evaluation of the field are upwards of R = .20, which is more convincing (and this is noted in the discussion).

As is now easily comparable (thanks to your suggestion regarding the standardized regression model and the semi-partial correlations), the effect we are most interested in, the interaction effect, is actually quite similar in size between all dependent variables in both studies. We agree, that these effects may appear small and that our models do not explain a lot of variance in total. However, we think our results can be considered relevant and we might even have underestimated the real-world impact of “me-search” for two reasons: 1) Our experimental manipulation was relatively subtle. 2) Participants were asked to judge the researchers on quite specific characteristics on the basis of very little information. We added a paragraph in the General Discussion dedicated to this aspect: “In general, the main models tested here explained between 5% and 28% of variance which may not appear impressive at first glance. However, our studies posed a very strict test of the effects of “me-search” by only using a subtle manipulation sparse in information followed by measures of very specific perceptions which might have contributed to an understatement of the real-world impact.” (p. 26).

In the Study 2 discussion, the authors suggest that “The moderation…persisted when controlling for self-identification as a vegan… this has to be interpreted carefully.” I suggest you remove this caution or ignore the effect: either interpret something or don’t. There is no carefulness when you report an effect with an accompanying p-value. If you are not convinced that this result is not a fluke then simply describe your data. Either way, please remove this claim of caution.

We removed the claim of caution (see p. 24).

Finally, you note that the expectations of neutrality was affected by the manipulation and claim this effect is “small” (d = .22). This is actually more of an average effect size in social/personality studies (see Funder & Ozer, 2019 and Gignac & Szodorai, 2016). I wouldn’t be so quick to dismiss it.

Additionally, earlier on in the manuscript (under Main effect of being affected) you interpret a similarly sized d = .25 at face value. If you have determined an alpha level of .05 as your criteria (which you did, and you also can see that smallest effect size based on the power analysis), you should stick to that criteria throughout and simply interpret the effects.

We removed the interpretation of the effect size, as the more important argument we want to make is the fact that adding neutrality expectations as control did not change the pattern of results: “Participants who read about the personally affected researcher had weaker expectations of neutrality; yet, when added to the regression model as a control, the pattern of results remained unchanged (see Appendix E in the supplementary materials, […]).” (p.25)

Finally, I admit that I am a bit puzzled by the overall pattern here: participants were less critical when Dr Lohr was affected; they saw results as more credible when Dr Lohr was affected; they trusted the results more when Dr Lohr was demonstrably biased?

What would happen if you asked the participant to rate whether the researcher is biased?

Thank you for these questions! In response, we analyzed the qualitative data from Study 2 and added these insights to the manuscript. Please, see our response to your last comment below.

General Discussion

This is again an issue in the general discussion: you claim that ‘when participant attitudes were less positive, this pattern flipped…” but this pattern did not flip. We simply see a reduction in strength.

Saying „the pattern flipped“, we wanted to express the fact that the conditional effect (mean between the conditions at +/-1 SD of attitudes towards the topic) changes its sign. We clarified that in our phrasing throughout both Results sections.

Conclusions

In sum, I think the key variable here is participant attitudes towards the topic. I’m not convinced of the moderation effect here for several reasons.

First, it does not reverse the direction of the effect: this is key for the claim

Second, the moderation itself is small. There’s not much of a difference between any two given slopes (with the exception of Figur 2a).

Third, the whole idea strikes me as strange: participant attitudes drive the effect always in a positive direction. Even when the researcher admits to being biased, participants don’t evaluate him more negatively. Plus, ‘bias’ itself wasn’t measures as far as I saw.

I think this is an interesting set of studies but it needs a bit more data to rule out some possible effects and I think the interpretation needs to be revised. I’d like to see these points addressed in a revision and I hope my comments are helpful in that regard.

References

Funder, D. C., & Ozer, D. J. (2019). Evaluating effect size in psychological research: Sense and nonsense. Advances in Methods and Practices in Psychological Science, 2(2), 156-168.

Gignac, G. E., & Szodorai, E. T. (2016). Effect size guidelines for individual differences researchers. Personality and individual differences, 102, 74-78.

Regarding your first and second conclusion: please, see Author Responses above. Regarding your third conclusion: We understand that it might appear odd at first glance that „me-search“ might have negative as well as positive effects on trust in science. However, we think that makes our two studies noteworthy. Following your question what – if asked directly – participants might say about an affected researcher, we decided to include the qualitative data we assessed in Study 2 (participants’ self-reported opinion on the researcher being affected) in the paper. Here, it becomes clear that most participants express a negative or at least mixed opinion about a researcher being affected, the argument of potential bias being mentioned most often in the open text responses. However, on average, the affected researchers in both studies were not judged more negatively than the not affected researchers. We added the analyses and interpretation of the qualitative data throughout the text: 

Study 2 – Method, p. 18: “[…] and an open-ended question about participants’ opinion regarding the researcher being personally affected to explore how laypeople rationalize their opinion. These responses were later coded for valence (positive, negative, mixed, or neutral) and content (deductive and inductive coding) by two raters blind to the specific research question (see Appendix C in the supplementary materials, https://osf.io/phfq3/?view_only=a3694575674944fababa32b696e6e645; interrater reliability for valence, Cohen’s κ=.86, p<.01; and for content, Cohen’s κ=.74, p<.01).”

Study 2 – Results, p. 23-24: “Participants’ Opinion. Overall, participants who responded to the open-ended question expressed mostly negative opinions about the researcher being affected by his own research (negative: 48%, neutral: 21%, positive: 17%, and mixed: 14%) about the researcher being affected by his own research. The most frequently mentioned (negative) remark was that a “me-searcher” might be biased towards their research (60%; e.g., “By introducing himself as being affected, I fear he cannot evaluate the results of his research objectively”). The second most frequently mentioned remark was that such idiosyncratic relevance is irrelevant (24%; e.g., “It wouldn’t make a difference”). Positive remarks were mentioned less frequently: Participants ascribed more motivation (11%; e.g. “I think interest, also personal interest, is an important prerequisite for determined research”) or knowledge about the topic (8%; e.g. “Very good, most likely, he thus is knowledgeable about the subject matter and can conduct the study in a more purposeful manner”) to the “me-searcher”, or recognized the transparency (7%; e.g., “The main thing is transparency. People are always biased, perhaps even unconsciously”; for more details, see Appendix C in the supplementary materials: https://osf.io/phfq3/?view_only=a3694575674944fababa32b696e6e645).”

Study 2 – Discussion, p. 24-25: “Further, qualitative analyses revealed that most participants reported negative – or, at least, mixed – perceptions of a “me-searcher” (e.g., “me-searchers” may be biased, but also highly motivation and knowledgeable), which corroborated our theoretical prediction that “me-search” may be a double-edged sword. Interestingly, these qualitative findings seem somewhat contradictory to the quantitative findings, according to which there was no main effect of researchers’ idiosyncratic affection by their research topic.”

General Discussion, p. 26: “’Me-search’ neither automatically sparks trust nor mistrust in laypeople, even if their explicit opinions seem rather negative. In line with assumptions from motivated science reception […]”

Reviewer #3: 

This paper examines how peoples’ attitudes toward a topic influence their evaluation of research and researchers’ trustworthiness - in the context of research topics which have personal relevance to the researchers.

I believe the aim of this study is very interesting and timely, and a great addition to the research; the study is well done. Also positive is the osf documentation and preregistration of study 2.

While I am quite enthusiastic about the paper, I also have a few issues to remark:

Abstract:

Reading your abstract for the first time, I was a bit confused whose attitudes and interests you were talking about when and what you were measuring – Maybe lead the reader straighter to the point, instead of mirroring your introduction (even though the intro read quite smoothly)?

Thank you for this feedback. We clarified the phrasing in the abstract.

Methods and Results:

• Please report the results of the factor analyses justifying the scales for testing your hypotheses, can be in the supplement as well.

We added the factor analyses in appendix A: See also p. 9 Factor analyses (see Appendix A in the supplementary materials, https://osf.io/phfq3/?view_only=a3694575674944fababa32b696e6e645) suggest that a two-factor model (with expertise and integrity/benevolence) fit the data better than a three-factor model (as suggested by [36]), corroborating the idea of a cognitive-rational dimension and an affective dimension of trustworthiness [37].”

• p values should be reported in accord with APA reporting guidelines, not p < .05.

We changed the display of our p values to exact values with three decimals in the text and to significance-indicators (*) of up to three decimals in the tables.

• Clearly state what you are testing when comparing the groups with attitudes +- 1SD from mean. “A regression was performed…” or as appropriate.

We performed simple slopes analyses and described the conditional effects (see response to Reviewer 2). This was not made clear enough, so we specified that in our phrasing in the manuscript throughout both Results sections.

• Table 1,2 and probably a question of taste: I don’t know of a convention in which Cronbach’s alphas are displayed in the diagonal of a correlation table. Since they’re conceptually something different than correlations, I would suggest moving them to a single column.

We changed the correlation tables (now Table 1 and 3) accordingly.

• I don’t see a theoretical reason why you should test trustworthiness, and then the two scales separately, would you please justify this? (I think this should be avoided, if there is no indication from theory, and since the results are interdependent anyway. Reminds me of MANOVA)

In this paper, we focused on overall trustworthiness and credibility. However, to gain further insights into the „building blocks of trust in science“ (see discussion on p. 27), we also explored whether and how the two facets of epistemic trustworthiness might be differentially impacted: While highly correlated, both might differentially impacted by “me-search” as there is reason to believe the two facets are, in fact, fueled by different perceptions (i.e., expertise by perceived competence and integrity/benevolence by more communal aspects). Thus, while not the focus of this paper, the separate analyses provide some tentative insights that are worth following up upon in future research (see also Author Response to the following concern).

Discussion (and maybe Introduction):

Finally, and probably most importantly, I am a bit unsure about your interpretation of the main effect on trustworthiness following the experimental variation – you discuss two ideas, and only in the “future research” section: a) there is a preference for anecdotal evidence vs. other types of evidence, i.e. someone affected has some type of special access to a research topic, and b) disclosing being affected / transparency signals benevolence (by the way, in line with intentionalist models of communication). Don’t your results – when dividing the trustworthiness measure into expertise and benevolence/integrity provide some (sure, provisional) evidence that allow you to dive deeper into these explanations in your argumentation? I believe your explanations of this effect could be argued more thoroughly. To that point, while I think it is alright that you only pose an exploratory research question in the introduction, here, you actually almost only provide anecdotal evidence yourselves to argue that question (which you may want to be clear about).

We fully agree with these assumptions and discuss these in the General Discussion in two paragraphs (i.e., regarding the effects on the two facets of epistemic trustworthiness and in the future research section). We tried to connect and refine these paragraphs, so that the argumentation becomes clearer (p. 26: “This points to “me-search” – when received positively – possibly adding to the perception of competence-related aspects like a deeper knowledge of a phenomenon (e.g., via anecdotal insights) [12–14] and, even more so, warmth-related aspects like seeming more sincere, benevolent, transparent and, thus, approachable [15,16,41].” and p. 27: “[…] but also important theoretical insights on the building blocks of trust in science and researchers (see discussion above regarding the effects on the facets of epistemic trustworthiness). As one example, knowing that a qualitative level of knowledge is highly valued could further research on the trust-benefit of enriching statistical evidence with anecdotal and narrative elements [44,45].”).

---

## [Editor Report · Decision Letter 1]

16 Jun 2021

When research is me-search: How researchers' motivation to pursue a topic affects laypeople's trust in science

PONE-D-20-38206R1

Dear Dr.  Altenmuller   

We’re pleased to inform you that your manuscript has been judged scientifically suitable for publication and will be formally accepted for publication once it meets all outstanding technical requirements.

Kind regards,

Lynn Jayne Frewer, MSc PhD

Academic Editor

PLOS ONE

---

## [Editor Report · Acceptance letter]

1 Jul 2021

PONE-D-20-38206R1 

When Research is Me-Search: How Researchers’ Motivation to Pursue a Topic Affects Laypeople’s Trust in Science 

Dear Dr. Altenmüller:

I'm pleased to inform you that your manuscript has been deemed suitable for publication in PLOS ONE. Congratulations! Your manuscript is now with our production department. 

Kind regards, 

on behalf of

Dr. Lynn Jayne Frewer 

Academic Editor

PLOS ONE